# Reduction in Heavy Rare Earth Diffusion Sources in Sintered Nd-Fe-B Magnets via Grain Boundary Diffusion of Dy_70_Ce_70−x_Cu_30_

**DOI:** 10.3390/ma17235784

**Published:** 2024-11-26

**Authors:** Dongbiao Yang, Xiao Gao, Shuwei Zhong, Honglong Yang, Farao Zhang

**Affiliations:** 1Jiangxi Provincial Key Laboratory of Magnetic Metallic Materials and Devices, College of Rare Earths, Jiangxi University of Science and Technology, Ganzhou 341000, Chinazfarao@materchem.com (F.Z.); 2Beilun Customs People’s Republic of China, Ningbo 315800, China; 3National Rare Earth Functional Materials Innovation Center, Ganzhou 341000, China

**Keywords:** Nd-Fe-B magnets, coercivity, permanent magnets, diffusion efficiency

## Abstract

This study investigates the effect of Ce on the diffusion behavior of Dy-Cu alloys. The addition of Ce reduces the diffusion source melting point and promotes the formation of low-melting alloy phases, benefiting the diffusion behavior. The diffusion source with 10 wt.% of Ce shows the best magnetic performance, the coercivity of the magnet increases from 18.47 kOe to 23.60 kOe, and the incremental coercivity reaches 5.13 kOe. Ce diffusion improves the utilization of Dy, enhances diffusion uniformity, and promotes coercivity improvement. Ce also optimizes and regulates grain boundary phase structure distribution, consistent with magnetic property changes. Dy_70_Cu_30_ with an excessive thick shell layer wastes Dy and reduces utilization, while Dy_60_Ce_10_Cu_30_ has a relatively thin and uniform shell layer. Ce mainly distributes in the intergranular phase region, promoting Dy diffusion from the intergranular phase to form a shell layer. Excessive Ce can distort the magnet’s crystal structure, hindering magnetic property improvement. This study provides insights into optimizing the diffusion process and improving the Dy-Cu alloy.

## 1. Introduction

Sintered Nd-Fe-B magnets have been widely used in various fields such as permanent magnet motors, wind power, and electronic products due to their excellent magnetic properties [1,2,3]. The magnetic properties of magnets at high temperatures are reduced by intrinsic properties, but the development of grain boundary diffusion technology (GBDP) has improved this problem [4,5,6,7]. This improvement is because the heavy rare earth (HRE) element diffusion via grain boundary (GB) can form a high magnetic anisotropy shell structure on the main phase [8,9], effectively suppressing the nucleation of reverse domains [10]. At the same time, the consistent distribution of GB phases can also reduce the cascade transfer in the demagnetization process, significantly improving the coercivity. Therefore, researchers have focused on how to improve the depth and efficiency of diffusion, such as the development of low-melting-point alloys [11,12,13,14,15], substrate design [16,17,18,19], and the improvement of diffusion technology [20,21,22,23,24], to enhance the utilization efficiency of HRE elements. With the continuous expansion of the back-end application market and the ever enhancing call for high-end magnets in the industry, the consumption of heavy rare earth diffusion sources has been rapidly increasing. Therefore, how to reduce the use of HRE such as Dy and Tb while ensuring the stability of magnet performance is a key issue in material development at this stage.

On the other hand, the Ce element, as the rare earth element with the largest reserves and the most widespread distribution, is facing a large backlog problem [25], which has led to limitations in its applications. Adding a single Ce element can lead to a sharp deterioration in the performance of Nd-Fe-B magnets, due to the reduction in their intrinsic magnetic properties and microstructure [26]. The dual-main phase preparation technology developed by Li et al. [27] can optimize the element distribution and improve the coercivity with high Ce content. This finding proves that the impact of Ce elements on magnets is not only negative. Therefore, introducing appropriate Ce elements into magnets can not only effectively improve magnetic properties but also reduce costs. Zhao et al. [28] found that Ce elements were uniformly distributed along the main phase edges through the grain boundary diffusion of TbCeLaCu alloys, which not only greatly improved the diffusion depth but also effectively reduced costs. Wang et al. [29] designed TbCeAlCuZn alloy diffusion sources, and the heat resistance of the magnets was significantly improved. However, the interaction between HRE elements and Ce elements in the diffusion process is still unclear and needs further exploration.

Replacing HRE elements in diffusion source alloys with Ce elements can not only improve the coercivity but also reduce manufacturing costs. This study is based on the DyCu alloy as the diffusion source, and a significant improvement of the diffusion effect is achieved by adding Ce to replace part of the Dy element. Further exploration of the synergistic effect of Ce element with other elements in the diffusion process provides experimental basis and theoretical guidance for the fabrication of high coercivity magnets.

## 2. Experimental Procedure

A commercial sintered magnet with 45 M grade manufactured by Ganzhou East Magnetic Rare Earth Co., Ltd (Ganzhou, China). was used as the diffusion source magnet. The substrates were obtained by a series of processes including melting, rapid solidification, hydrogen decrepitation, jet milling, orientation pressing, sintering, and annealing, and then they were cut into 10 × 10 × 5 mm^3^ square samples using an electrical discharge wire cutting machine. The diffusion source alloy was composed of Dy_x_Ce_70−x_Cu_30_ (x = 70/60/50/40/30), and the alloy raw materials were pure Dy (99.9%), pure Ce (99.9%), and pure Cu (99.95%) metals produced by Zhongnuo New Materials Technology Co., Ltd (Beijing, China). The raw materials were placed into a high vacuum arc melting device and melted 5 times to obtain a homogeneous metal ingot. A ribbon-shaped thin strip was obtained using a high vacuum rapid quenching strip caster at a copper roller speed of 25 m/s. The obtained metal thin strip was cut into pieces and then uniformly attached to the ground and the cleaned Nd-Fe-B original magnet for tape casting treatment, with the tape weight accounting for 0.7 wt.% of the magnet mass. The samples were placed in a high-vacuum diffusion furnace for diffusion treatment, with a diffusion treatment process of 900 °C × 6 h and 500 °C × 3 h. The diffusion magnet was obtained after cooling down with the furnace.

The phase transition temperature of the diffusion source alloy was measured using a simultaneous thermal analyzer in the temperature range of 25~1000 °C, with a heating rate of 5 °C/min. The magnetic behavior of the magnet at room temperature, both before and after the diffusion, was evaluated using a NIM-500C high-temperature permanent magnetometer (Beijing, China). Changes in the crystal structure were analyzed through X-Ray diffraction (XRD) using a Panalytical Empyrean system (Malvern, UK). The magnet’s microstructure was examined with a Scanning Electron Microscope (SEM), model MIRA3 LMH (Kohoutovice, Czech Republic). To assess the depth of diffusion within the magnets, an Electron Probe Micro-Analyzer (EPMA), specifically the JXA-iSP100 model, was employed (Tokyo, Japan).

## 3. Results and Discussion

To investigate the influence of a Ce element on the melting point (MP) of the diffusion source alloy, the thermal flow curve was measured. Figure 1 shows the DSC curves of the diffusion source alloy with different compositions. The Dy_70_Cu_30_ alloy exhibits a sharp endothermic peak at 863.8 °C, corresponding to the MP of the alloy. When the Ce content is 10 wt.%, its melting point decreases to 823.2 °C. This result indicates that the addition of a Ce element can lower the MP of the diffusion source alloy. As the Ce content further increases to 20 wt.%, 30 wt.%, and 40 wt.%, more than two endothermic peaks appear in each group of curves, and their peak values are all lower than the melting point of the Dy_70_Cu_30_ alloy. This finding suggests that the increase in Ce content not only lowers the MP of the diffusion source but also promotes the formation of multiple low-melting alloy phases, which is beneficial for the occurrence of diffusion behavior.

The second quadrant M-H curves of the Dy_x_Ce_70−x_Cu_30_ (x = 70/60/50/40/30) diffusion are shown in Figure 2. The corresponding increments in magnetic properties and coercivity (*H*_cj_) are presented in Table 1. As depicted in Figure 2a, the coercivity of the Dy_70_Cu_30_ diffused magnet increased from 18.47 kOe to 23.47 kOe, while the remanence (*B*_r_) and magnetic energy density ((BH)_max_) reached 13.15 kGs and 43.17 MGOe, respectively. The magnet diffused with the Dy_60_Ce_10_Cu_30_ alloy exhibiting a *H*_cj_ increment of 5.13 kOe, with no significant decrease in *B*_r_ or ((BH)_max_), and an improvement in squareness. This finding indicates that increasing the Ce content can not only effectively reduce the consumption of Dy, but also improve the uniformity of diffusion. As the Ce content further increased to 20 wt.%, 30 wt.%, and 40 wt.%, the increments in coercivity decreased successively to 4.69 kOe, 3.68 kOe, and 3.26 kOe, respectively. It is important to note that, as the Ce content in the diffusion source increased, the squareness of the magnet also improved significantly. This finding suggests that the diffusion of Ce elements can not only promote the improvement of coercivity but also has a certain effect on uniformity improvement.

Figure 3 depicts the XRD curves of the initial magnet and the magnets diffused with different alloys. As can be observed, the diffraction peaks of the initial magnet are dominantly Nd_2_Fe_14_B, and no new phase is generated in the magnet diffused with the Dy_70_Cu_30_ alloy. Analysis shows that the Nd_2_Fe_14_B diffraction peak of the diffused magnet shifts towards larger angles, indicating that the Dy element enters the main phase and causes crystal lattice to shrink [30]. The magnet diffused with the Dy_60_Ce_10_Cu_30_ alloy exhibited new phases such as Nd_2_O_3_ and CeO, indicating that the diffusion of the alloy can cause changes in the crystal structure. Moreover, with the Ce content in the diffusion source increased, the composition and proportion of the phase structure became more complex. This finding indicates that an appropriate amount of the Ce element can improve the composition of the crystal structure, while too much Ce can cause distortion of the crystal structure of the magnet, which is not conducive to further magnetic property improvement.

Figure 4 depicts the backscattered scanning electron microscopy micrographs of the original magnet and the magnets diffused with different diffusion sources at various depths from the diffusion surface. Rows a–f show the original magnet and the magnets diffused with Dy_70_Cu_30_, Dy_60_Ce_10_Cu_30_, Dy_50_Ce_20_Cu_30_, Dy_40_Ce_30_Cu_30_, and Dy_30_Ce_40_Cu_30_, respectively. The dark gray areas illustrates the main phase, and the bright white areas represent the GB phase. Except for the original magnet, the backscattered images of the Dy_60_Ce_10_Cu_30_ diffused magnets clearly show a gray-shaded region at the outer layer of the main phase at a depth of 30 μm, which is the shell structure of (Pr, Nd, Dy)_2_Fe_14_B formed by the Dy element replacing the Pr and Nd atoms in the main phase outer layer during the diffusion process, significantly improving the *H*_cj_. Moreover, it is observed that the magnets diffused with alloys have more small-sized grains, indicating that the Ce element in the diffusion source plays a role in refining the grain size. The addition of the Ce element also inhibits the formation of sharp regions in the main phase grains, effectively limiting the formation of antiferromagnetic domains. The GB phase in the diffused magnets is more abundant, forming a continuous thin layer, which effectively isolates the main phase and enhances the *H*_cj_ of the magnet. Therefore, it can be inferred that the Ce element can optimize and regulate the distribution of the GB phase structure, which is in agreement with the change in magnetic properties.

By analyzing the backscattered micrographs of the Dy_70_Cu_30_ and Dy_60_Ce_10_Cu_30_ diffused magnets at a depth of 100 μm, it can be observed that both the magnets have a core–shell structure (Figure 5). However, it is observed that the shell of the Dy_70_Cu_30_ diffused magnet is very thick, with many shells occupying half or even more than two-thirds of the entire grain. Studies have shown that, when the shell thickness exceeds 15 nm, the improvement in coercivity is limited. Therefore, the excessively thick shells in the Dy_70_Cu_30_ diffused magnet leads to a low utilization rate of the Dy element. The shell of the Dy_60_Ce_10_Cu_30_ diffused magnet is relatively thin and uniform, and no excessively thick shell is observed in the backscattered images of the Dy_50_Ce_20_Cu_30_, Dy_40_Ce_30_Cu_30_, and Dy_30_Ce_40_Cu_30_ diffused magnets. This indicates that the Ce element can optimize the core–shell structure.

The distribution depth of the diffusion source is an important factor in evaluating its diffusion efficiency. The better the diffusion efficiency, the deeper the diffusion source penetrates into the interior of the magnet, thereby forming more core–shell structures and improving the utilization of Dy. Figure 6 shows the EPMA distribution maps of Dy selected at a depth of 1000 μm for Dy_70_Cu_30_, Dy_60_Ce_10_Cu_30_, and Dy_30_Ce_40_Cu_30_ magnets after diffusion. Interestingly, among all diffusion magnets, the Dy_70_Cu_30_ diffusion magnet had the deepest diffusion depth of Dy elements, which is related to its highest amount of Dy used. According to the EPMA maps of Dy element distribution, it can be seen that the Dy_70_Cu_30_ diffusion magnet has a large content of Dy elements enriched in the shallow layer, which also confirms that a thick Dy-rich shell layer was formed at a depth of 100 μm from the diffusion surface in the backscattering map of the Dy_70_Cu_30_ diffusion magnet, indicating why the Dy element diffused the deepest in the Dy_70_Cu_30_ diffusion magnet, but its coercivity was not as good as that of the Dy_60_Ce_10_Cu_30_ diffusion magnet. This finding suggests that Ce elements can optimize the morphology of the Dy-rich shell layer.

Figure 7 shows the EPMA distribution maps of the Ce element in the depth range of 100 μm after the diffusion of Dy_60_Ce_10_Cu_30_ and Dy_30_Ce_40_Cu_30_ magnets. When the Ce content is 10 wt.%, Ce elements are distributed relatively evenly in the magnet. When the Ce content increases to 40 wt.%, Ce elements accumulate in the surface region and are difficult to penetrate into the magnet interior. The distribution status of Dy and Ce is similar, exhibiting a uniform distribution locally. Therefore, we have revised our statement on this aspect. Additionally, we also posit that Dy and Ce undergo cooperative diffusion at elevated temperatures. As depicted in Figure 7a, when the Ce content is at 10%, the distribution of Dy and Ce is nearly identical. With an increase in Ce content to 40%, despite noticeable disparities, there is still evidence of cooperative diffusion, albeit with reduced uniformity in distribution (Figure 7b). This result indicates that, at a lower Ce content, there is a mutual promotion effect of diffusion between Dy and Ce elements. However, as the Ce content increases, it hinders the deep-level diffusion of Dy elements.

To further investigate the Dy and Ce distribution in the magnet crystal phase, surface scans were performed at a diffusion depth of 150 μm from the diffusion surface using a wavelength dispersive spectrometer (WDS). Figure 8 shows the WDS surface scan maps. Rows a, b, and c represent the Dy_70_Cu_30_, Dy_60_Ce_10_Cu_30_, and Dy_30_Ce_40_Cu_30_ diffusion magnets, respectively. Firstly, observing the distribution of Dy in the Dy_70_Cu_30_ diffusion magnet, it can be found that Dy elements are mainly distributed in the Dy-rich shell layer. However, the Dy-rich shell layer occupies a large area of the grain, which is consistent with the excessively thick shell layer found in the SEM analysis earlier, indicating that the core–shell structure of the Dy_70_Cu_30_ diffusion magnet is not ideal, and a lot of Dy elements are wasted. The formation of such grains not only wastes a lot of Dy and reduces its utilization rate but also reduces the volume fraction of the Nd_2_Fe_14_B phase, which is partially converted into the Dy_2_Fe_14_B phase. This is also the main reason why the coercivity and remanence ratio of the Dy_70_Cu_30_ diffusion magnet are lower than those of the Dy_60_Ce_10_Cu_30_ diffusion magnet. Observing the distribution of Dy in the Dy_60_Ce_10_Cu_30_ and Dy_30_Ce_40_Cu_30_ diffusion magnets, it can be found that Dy is uniformly distributed in the Dy-rich shell layer, and the shell thickness is moderate, which leads to a higher utilization rate of Dy. At the same time, it was found that Ce elements are mainly distributed in the intergranular phase, preferentially enriched in the GB region, and a small content of Ce enters the main phase region, which can effectively optimize the core–shell structure.

Continuing with the surface scans of the three diffusion magnets, four points were selected on the backscattering map for each magnet, and point scans were performed to quantitatively analyze the elements in the corresponding regions, as shown in Table 2. For the Dy_70_Cu_30_ and Dy_60_Ce_10_Cu_30_ diffusion magnets, point 1 represents the shell layer, point 2 represents the main phase core region, and points 3 and 4 represent the intergranular phase. For the Dy_30_Ce_40_Cu_30_ diffusion magnet, point 1 represents the main phase core region, point 2 represents the shell layer, and points 3 and 4 represent the intergranular phase. The Dy is relatively abundant in the shell layer and intergranular phase. This is because, during intergranular diffusion, Dy elements enter the interior of the magnet through the grain boundaries and form an epitaxial Dy-rich shell layer on the surface of the grains under the influence of concentration gradients and chemical binding energies. The ratio of Dy between the shell layer and intergranular phase in the Dy_70_Cu_30_ diffusion magnet is about 1.05, and, in the Dy_60_Ce_10_Cu_30_ diffusion magnet, it is about 2, while, in the Dy_30_Ce_40_Cu_30_ diffusion magnet, it is about 4.27. It can be observed that, with the increment of Ce in the diffusion source, the ratio of Dy element content between the shell layer and intergranular phase in the corresponding diffusion magnets gradually increases.

This finding indicates that Ce elements can promote the diffusion of Dy elements from the intergranular phase to the interior of the grains to form a shell layer. This is mainly because the chemical binding energy of Ce elements in the intergranular phase is higher than that of Dy elements, which promote the diffusion of Dy elements towards the interior of the grains. An analysis of the Ce element content shows that it is mainly distributed in the intergranular phase region. When the diffusion source lacks Ce, the Cu element predominantly localizes within the primary phase, while the Dy element establishes a robust Dy-rich shell alongside the epitaxial layer of the primary phase. In cases where the diffusion source includes 10% Ce, a significant portion of Ce disperses within the grain boundary phase rather than infiltrating the primary phase, thereby mitigating intrinsic magnetism and enhancing the fluidity of the grain boundary phase, leading to optimized microstructure. Conversely, excessive Ce content (40%) results in an upsurge in Ce concentration within the primary phase, subsequently diminishing the intrinsic characteristics.

## 4. Conclusions

The effect of a Ce element on the diffusion behavior of the Dy-Cu alloy was investigated. The addition of a Ce element was found to lower the diffusion source melting point and promote the formation of multiple low-melting alloy phases, which is beneficial for diffusion behavior. The diffusion source with 10 wt.% of Ce shows the best magnetic performance. The coercivity of the magnet increases from 18.47 kOe to 23.60 kOe, and the incremental coercivity reaches 5.13 kOe. The diffusion of the Ce element not only improves the utilization of Dy but also enhances the uniformity of diffusion and promotes the improvement of coercivity. The Ce element can also optimize and regulate the distribution of the grain boundary phase structure, which is consistent with the change in magnetic properties. The excessive thick shell layer in the Dy_70_Cu_30_ diffusion magnet wastes a lot of Dy and reduces the utilization rate of Dy, while the Dy_60_Ce_10_Cu_30_ diffusion magnet has a relatively thin and uniform shell layer. Ce elements are mainly distributed in the intergranular phase region, and their addition can promote the diffusion of Dy elements from the intergranular phase to the interior of the grains to form a shell layer. The results show that the right amount of the Ce element can improve the composition of the crystal structure, while too much Ce can cause distortion of the crystal structure of the magnet, which is not conducive to further the improvement of magnetic properties. This study provides some insights into the optimization of the diffusion process and the improvement of Dy-Cu alloys.

## Figures and Tables

**Figure 1 materials-17-05784-f001:**
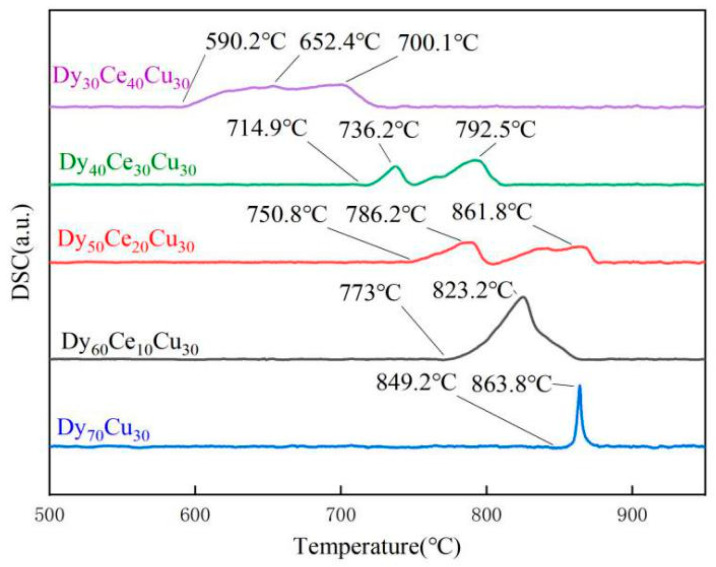
The DSC curves of the diffusion source alloy with different compositions.

**Figure 2 materials-17-05784-f002:**
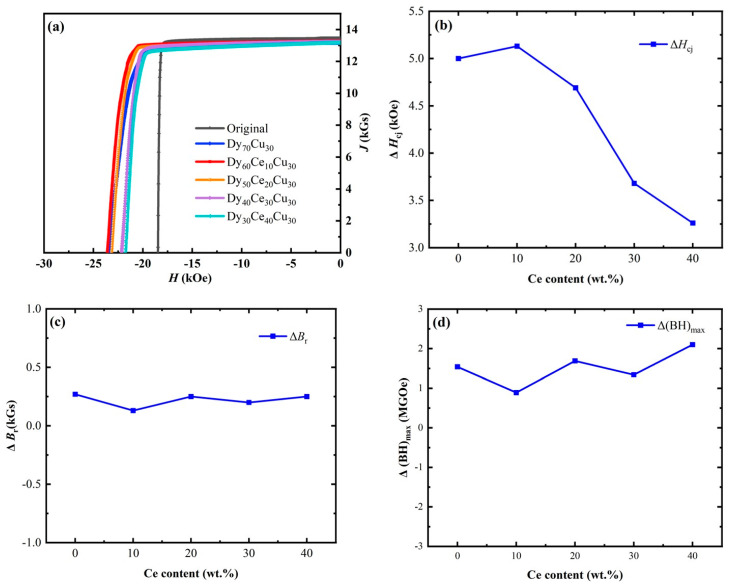
(**a**) The second quadrant M-H curves of the magnets pre and post Dy_x_Ce_70−x_Cu_30_ (x = 70/60/50/40/30) diffusion; the magnetic properties of the magnet were studied as a function of Ce content in the diffusion source alloy; (**b**) the variation of *H*_cj_ (Δ*H*_cj_); (**c**) the variation of *B*_r_ (Δ*B*_r_); (**d**) and the variation in (BH)_max_ (Δ(BH)_max_).

**Figure 3 materials-17-05784-f003:**
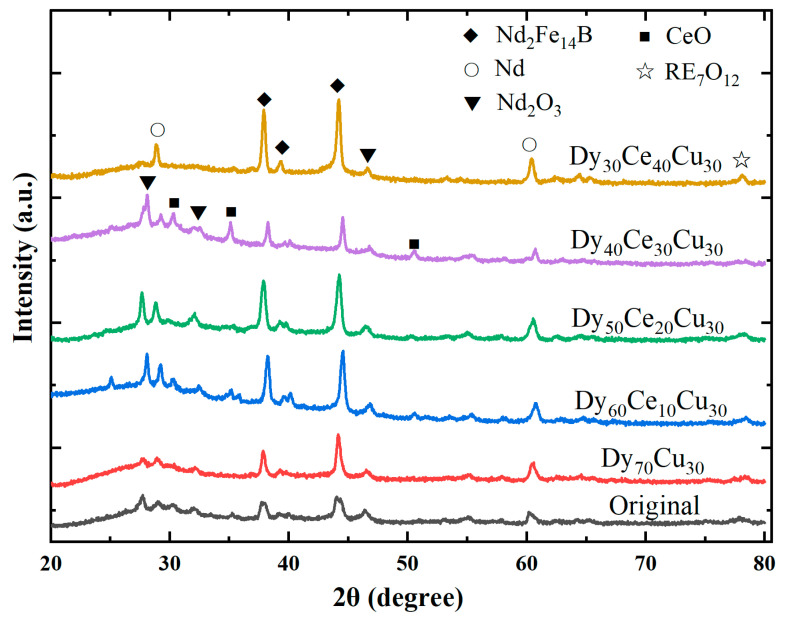
The XRD curves of the initial magnet and the magnet after diffusion with different diffusion sources.

**Figure 4 materials-17-05784-f004:**
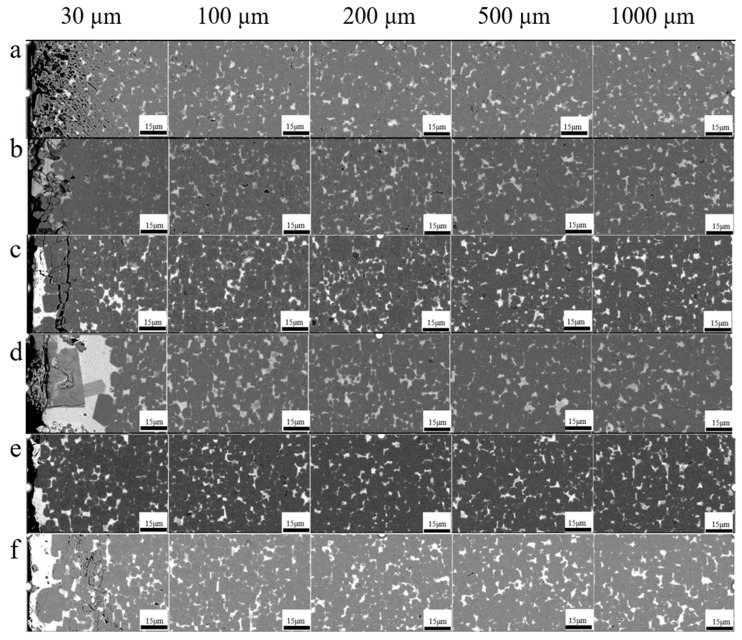
The backscattered scanning electron microscopy images of the original magnet and the magnets diffused with different diffusion sources at depths of 30 μm, 100 μm, 200 μm, 500 μm, and 1000 μm from the diffusion surface. Figure a–f is as follows: (**a**) original magnet, (**b**) Dy_70_Cu_30_ diffused magnet, (**c**) Dy_60_Ce_10_Cu_30_ diffused magnet, (**d**) Dy_50_Ce_20_Cu_30_ diffused magnet, (**e**) Dy_40_Ce_30_Cu_30_ diffused magnet, and (**f**) Dy_30_Ce_40_Cu_30_ diffused magnet.

**Figure 5 materials-17-05784-f005:**
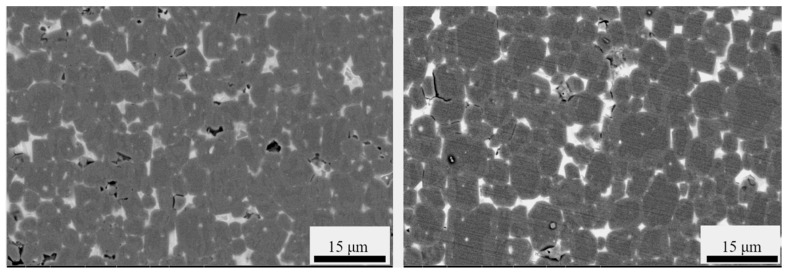
The backscattered images of the Dy_70_Cu_30_ and Dy_60_Ce_10_Cu_30_ diffused magnets at a depth of 100 μm from the diffusion surface are presented in this study.

**Figure 6 materials-17-05784-f006:**
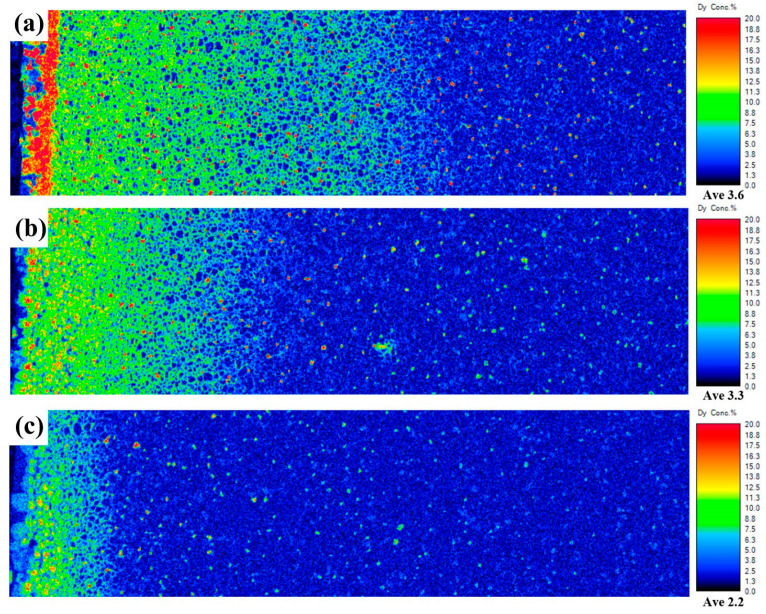
The EPMA distribution maps of the Dy element in the depth range of 0–1000 μm after the diffusion of (**a**) Dy_70_Cu_30_, (**b**) Dy_60_Ce_10_Cu_30_, and (**c**) Dy_30_Ce_40_Cu_30_ magnets.

**Figure 7 materials-17-05784-f007:**
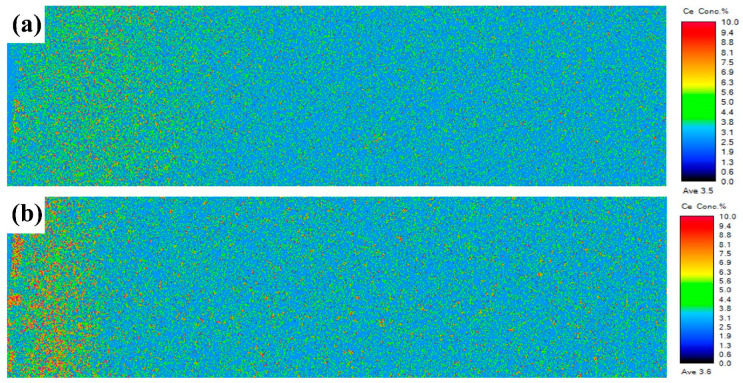
The EPMA distribution maps of the Ce element in the depth range of 100 μm after the diffusion of (**a**) Dy_60_Ce_10_Cu_30_ and (**b**) Dy_30_Ce_40_Cu_30_ magnets.

**Figure 8 materials-17-05784-f008:**
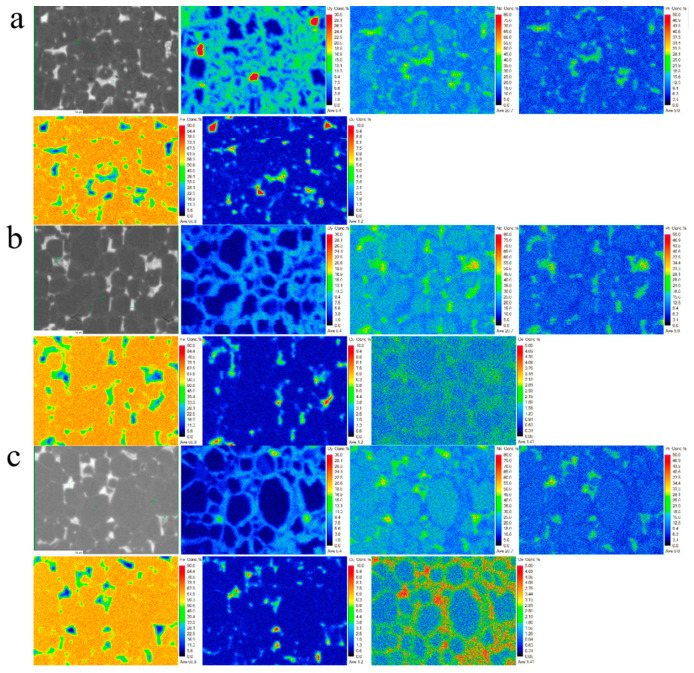
The EPMA mapping of the microscopic region of (**a**) the Dy_70_Cu_30_ diffused magnet, (**b**) the Dy_60_Ce_10_Cu_30_ diffused magnet, and (**c**) the Dy_30_Ce_40_Cu_30_ diffused magnet at a depth of ~100 μm after the diffusion.

**Table 1 materials-17-05784-t001:** The magnetic properties of the selected magnets.

Samples	*H*_cj_ (kOe)	*B*_r_ (kGs)	(BH)_max_ (MGOe)	*H*_k_/*H*_cj_ (%)	Δ*H*_cj_ (kOe)
Original	18.47	13.42	44.71	98.6	-
Dy_70_Cu_30_ diffusion	23.47	13.15	43.17	87.4	5.00
Dy_60_Ce_10_Cu_30_ diffusion	23.60	13.29	43.82	91.1	5.13
Dy_50_Ce_20_Cu_30_ diffusion	23.16	13.17	43.02	91.3	4.69
Dy_40_Ce_30_Cu_30_ diffusion	22.15	13.22	43.37	92.3	3.68
Dy_30_Ce_40_Cu_30_ diffusion	3.26	13.18	42.60	92.2	3.26

**Table 2 materials-17-05784-t002:** The EDS results of diffused elements in magnetic regions of Dy_70_Cu_30_, Dy_60_Ce_10_Cu_30_, and Dy_30_Ce_40_Cu_30_ magnets.

Diffused Magents		Ce	Dy	Cu	Co	Pr	Nd	Fe	Al	Ga	B	O
Dy_70_Cu_30_	1	-	3.73	0.00	1.46	1.87	5.82	76.16	1.11	0.04	7.83	1.97
	2	-	0.17	0.04	1.62	2.51	8.60	77.86	1.36	0.09	5.79	1.96
	3	-	2.51	6.12	3.72	6.93	15.74	53.04	1.18	1.50	6.10	3.17
	4	-	2.59	7.06	4.15	10.92	25.30	36.29	3.92	2.88	0.20	6.69
Dy_60_Ce_10_Cu_30_	1	0.13	2.82	0.02	1.63	1.82	7.15	76.62	1.19	0.08	6.08	2.46
	2	0.02	0.13	0.11	1.69	2.44	8.61	74.99	1.19	0.08	8.58	2.16
	3	0.36	1.42	6.16	6.83	11.51	25.88	35.21	1.33	6.30	0.00	5.00
	4	0.21	1.41	0.19	0.95	6.06	12.58	43.82	0.34	0.05	2.37	32.02
Dy_30_Ce_40_Cu_30_	1	0.03	0.12	0.05	1.73	2.65	8.68	78.71	1.28	0.14	6.61	0.00
	2	0.41	2.22	0.02	1.56	2.07	7.17	77.92	1.42	0.00	7.21	0.00
	3	2.53	0.52	6.35	4.42	15.81	31.44	26.10	2.16	2.20	2.18	6.29
	4	0.72	6.20	0.00	0.50	6.35	20.81	15.22	0.17	0.02	0.00	50.01

## Data Availability

The original contributions presented in the study are included in the article, further inquiries can be directed to the corresponding author.

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
