# Peer review of "Reduction in Heavy Rare Earth Diffusion Sources in Sintered Nd-Fe-B Magnets via Grain Boundary Diffusion of Dy70Ce70−xCu30"

_materials, 2024, doi:10.3390/ma17235784_

Round 1

Reviewer 1 Report

Comments and Suggestions for Authors

The authors investigated the effect of Dy diffusion on the magnetic properties of NdFeB sintered magnets. The diffusion sources were Dy(x)Cu(70-x)Ce(30) (x=70, 60, 50, 40, 30) alloys. The study revealed that the addition of Ce reduces the melting temperature of the grain boundary phase, facilitating the efficient diffusion of Dy and enhancing coercivity. It was observed that Dy diffusion was particularly effective when the source contained a small amount of Ce (10 wt%). While the experiments were conducted systematically, the manuscript requires revisions as outlined below. The authors should address the following queries before its consideration for publication in MDPI-Materials.

1. In the abstract, the authors should describe how much the magnetic properties are improved by diffusion process. I think the diffucion source with 10 wt% Ce shows the best magnetic performance. It shold be also written in Conclusion. 

2. In page 3, the authros say 'As the Ce content in the diffusion source further increased to 20 wt.%, 30 wt.%, and 40 wt.%, the increments in coercivity decreased successively to 4.69 kOe, 3.68 kOe, and 3.26 kOe, respectively.'. Why this is your success? The coercivity reduced. 

3. At page 5 line 11, the authors mention that 'the backscattered images of the other magnets clearly show a gray-shaded region at the outer layer of the main phase'. But I do not find where it is. Clarify by arrows or schematics. 

4. The authors also say that 'it is observed that the magnets diffused with alloys have more small-sized grains'. Indicate where it is. 

5. At the last line in page 5, the authors say that 'magnet has core-shell strucutre'. I do not find it neither. 

6. Caption in Figure 4: The authors shold written more information. What are a-f?

7. Caption in Figure 5: the authors shold descrive (a) and (b) are duplicate from images in Figure 4. 

8. In page 6, the authors describe the difference of two SEM images in Figure 5. Dy shell is not clear. Try adjust the gray scale or what ever. 

9. Figure 8 and the explanation for that in page 9 are not understandable. 

10. Figure captions are not sufficiently written through out the article, in particular Figure 4, 5, 6, 7, and 8. EPMA images are easy to visualze the diffusion, however explanations are not sufficient and results are unclear. 

11. Explanations for Table 2 are not sufficient. Where are the points?

12. page3, line9: Figure 1(a) -> Figure 2(a)

13. Comments on Figure 7. Although the authors stated that Ce is distributed almost homogeneously within the magnets, it appears to be denser near the surface in (a) and (b). In my view, these distributions closely resemble the Dy diffusion patterns shown in Figure 6 (b) and (c). Therefore, it looks like Dy and Ce are diffused together.

Author Response

Reviewer 1: The authors investigated the effect of Dy diffusion on the magnetic properties of NdFeB sintered magnets. The diffusion sources were Dy(x)Cu(70-x)Ce(30) (x=70, 60, 50, 40, 30) alloys. The study revealed that the addition of Ce reduces the melting temperature of the grain boundary phase, facilitating the efficient diffusion of Dy and enhancing coercivity. It was observed that Dy diffu sion was particularly effective when the source contained a small amount of Ce (10 wt%). While the experiments were conducted systematically, the manuscript requires revisions as outlined below. The authors should address the following queries before its consideration for publication in MDPI-Materials.

Response: Thank you for your valuable feedback on our research. We have carefully addressed your comments and made necessary revisions to the manuscript, including a detailed explanation of Dy diffusion's specific impact on magnetic properties, enhanced descriptions of the experimental methods, supporting data regarding the effect of Ce content on melting point, and an in-depth analysis in the discussion section. The modified parts are highlighted in red for your convenience. We appreciate your guidance in improving our paper and look forward to any further suggestions you may have.

Comment 1: In the abstract, the authors should describe how much the magnetic properties are improved by diffusion process. I think the diffucion source with 10 wt% Ce shows the best magnetic performance. It shold be also written in Conclusion.

Response: Thank you for your suggestion. Magnetic properties have been added to the abstract and conclusion.

The diffucion source with 10 wt% Ce shows the best magnetic performance, the coercivity of the magnet increases from 18.47 kOe to 23.60 kOe, and the incremental coercivity reaches 5.13 kOe.

Comment 2: In page 3, the authros say 'As the Ce content in the diffusion source further increased to 20 wt.%, 30 wt.%, and 40 wt.%, the increments in coercivity decreased successively to 4.69 kOe, 3.68 kOe, and 3.26 kOe, respectively.'. Why this is your success? The coercivity reduced.

Response: Thank you for your careful reading. "As the Ce content in the diffusion source further increased to 20 wt.%, 30 wt.%, and 40 wt.%, the increments in coercivity decreased successively to 4.69 kOe, 3.68 kOe, and 3.26 kOe, respectively." This indicates that although the increase in coercivity diminished, the performance was still optimized. When the Ce content was at 10%, the increase in coercivity was maximized, demonstrating that the magnetic properties did not deteriorate.

Comment 3: At page 5 line 11, the authors mention that 'the backscattered images of the other magnets clearly show a gray-shaded region at the outer layer of the main phase'. But I do not find where it is. Clarify by arrows or schematics.

Response: We have corrected this. Only Fig. c (Dy60Ce10Cu30 diffused magnet) show a gray-shaded region at the outer layer of the main phase at 30 μm.

Comment 4: The authors also say that 'it is observed that the magnets diffused with alloys have more small-sized grains'. Indicate where it is. 

Response: By comparing Figures 4b and 4c, it can be observed that the main phase grains in Fig. 4b exhibit a blocky distribution, indicating a larger grain size distribution. In contrast, Fig. 4c shows a higher proportion of grain boundary phases, which effectively isolates the main phase grains, resulting in smaller grain sizes.

Comment 5: At the last line in page 5, the authors say that 'magnet has core-shell strucutre'. I do not find it neither. 

Response: As shown in Figure 1, there are regions with varying contrast within the same main phase grain. The central region is referred to as the core layer, while the outer region is known as the shell layer, thus defining this structure as a core-shell structure.

Fig. 1

Comment 6: Caption in Figure 4: The authors shold written more information. What are a-f?

Response: Thank you for your valuable advice. We have added more information to Fig. 4.

Fig. a-f is as follows: a. Original magnet, b. Dy70Cu30 diffused magnet, c. Dy60Ce10Cu30 diffused magnet, d. Dy50Ce20Cu30 diffused magnet, e. Dy40Ce30Cu30 diffused magnet, f. Dy30Ce40Cu30 diffused magnet.

Comment 7: Caption in Figure 5: the authors shold descrive (a) and (b) are duplicate from images in Figure 4. 

Response: Yes, we analyzed the microscopic structures in the picture in further detail. This is a good help for readers to understand the difference between the magnetic properties.

Comment 8: In page 6, the authors describe the difference of two SEM images in Figure 5. Dy shell is not clear. Try adjust the gray scale or what ever. 

Response: The images in this article have been revised, as shown in Fig 2.

Fig. 2

Comment 9: Figure 8 and the explanation for that in page 9 are not understandable. 

Response: The corresponding explanation has been improved.

Comment 10: Figure captions are not sufficiently written through out the article, in particular Figure 4, 5, 6, 7, and 8. EPMA images are easy to visualze the diffusion, however explanations are not sufficient and results are unclear. 

Response: Thank you, we have added the figure captions for Figure 4, 5, 6, 7, and 8 as follows:

Fig. 4. The backscattered scanning electron microscopy images of the original magnet and the magnets diffused with different diffusion sources at depths of 30 μm, 100 μm, 200 μm, 500 μm, and 1000 μm from the diffusion surface. Fig. a-f is as follows: a. Original magnet, b. Dy70Cu30 diffused magnet, c. Dy60Ce10Cu30 diffused magnet, d. Dy50Ce20Cu30 diffused magnet, e. Dy40Ce30Cu30 diffused magnet, f. Dy30Ce40Cu30 diffused magnet.

Fig. 5. The backscattered images of the Dy70Cu30 and Dy60Ce10Cu30 diffused magnets at a depth of 100 μm from the diffusion surface are presented in this study.

Fig. 6. The EPMA distribution maps of Dy element in the depth range of 0-1000 μm after diffusion of (a) Dy70Cu30, (b) Dy60Ce10Cu30 and (c) Dy30Ce40Cu30 magnets.

Fig. 7. The EPMA distribution maps of Ce element in the depth range of 100 μm after diffusion of (a) Dy60Ce10Cu30 and (b) Dy30Ce40Cu30 magnets.

Fig.8. The EPMA mapping of microscopic region of (a) Dy70Cu30 diffused magnet, (b) Dy60Ce10Cu30 diffused magnet and (c) Dy30Ce40Cu30 diffused magnet at the depth of ~100 μm after the diffusion.

Fig. 8 illustrates the elemental distribution at a depth of approximately 100 μm following the diffusion of magnets utilizing various diffusion sources. The primary objective is to examine the distribution of the core-shell structure post full diffusion, facilitating efficient diffusion processes. A comprehensive explanation and analysis of this aspect are also provided in detail.

When the diffusion source lacks Ce, the Cu element predominantly localizes within the primary phase, while the Dy element establishes a robust Dy-rich shell alongside the epitaxial layer of the primary phase. In cases where the diffusion source includes 10 % Ce, a significant portion of Ce disperses within the grain boundary phase rather than infiltrating the primary phase, thereby mitigating intrinsic magnetism and enhancing the fluidity of the grain boundary phase, leading to optimized microstructure. Conversely, excessive Ce content (40 %) results in an upsurge in Ce concentration within the primary phase, subsequently diminishing the intrinsic characteristics.

Comment 11: Explanations for Table 2 are not sufficient. Where are the points?

Response: The EDS results in Table 2 comes from the backscattered image of Fig. 8, and the image has been modified. The corresponding explanation has been improved.

When the diffusion source lacks Ce, the Cu element predominantly localizes within the primary phase, while the Dy element establishes a robust Dy-rich shell alongside the epitaxial layer of the primary phase. In cases where the diffusion source includes 10% Ce, a significant portion of Ce disperses within the grain boundary phase rather than infiltrating the primary phase, thereby mitigating intrinsic magnetism and enhancing the fluidity of the grain boundary phase, leading to optimized microstructure. Conversely, excessive Ce content (40%) results in an upsurge in Ce concentration within the primary phase, subsequently diminishing the intrinsic characteristics.

Comment 12: page3, line9: Figure 1(a) -> Figure 2(a)

Response: This mistake had been corrected.

Comment 13: Comments on Figure 7. Although the authors stated that Ce is distributed almost homogeneously within the magnets, it appears to be denser near the surface in (a) and (b). In my view, these distributions closely resemble the Dy diffusion patterns shown in Figure 6 (b) and (c). Therefore, it looks like Dy and Ce are diffused together.

Response: Yes, your perspective aligns with ours. The distribution status of Dy and Ce is similar, exhibiting a uniform distribution locally. Therefore, we have revised our statement on this aspect. Additionally, we also posit that Dy and Ce undergo cooperative diffusion at elevated temperatures. As depicted in Fig. (a), when the Ce content is at 10 %, the distribution of Dy and Ce is nearly identical. With an increase in Ce content to 40 %, despite noticeable disparities, there is still evidence of cooperative diffusion, albeit with reduced uniformity in distribution (Fig. (b)).

Reviewer 2 Report

Comments and Suggestions for Authors

The manuscript “Reduction of Heavy Rare Earth Diffusion Sources in Sintered Nd-Fe-B Magnets via Grain Boundary Diffusion of Dy70Ce70-xCu30” provides important scientific results in the field of magnetic materials, particularly on the diffusion behavior of Dy-Cu alloys with Ce addition. It incorporates all relevant references and provides detailed reasoning and characterization.

However, upon reviewing the manuscript, I found some format editing is required. Please proofread and check the format and subscripts of the composition, for example, on page 3 in the last paragraph, and ensure the font size and subscripts of the compound composition are consistent. Adjusting these elements will improve readability and the overall quality of the manuscript.

 The research addresses how the addition of cerium (Ce) influences the diffusion behavior and overall properties of dysprosium-copper (Dy-Cu) alloys. The study specifically explores how Ce affects melting points, diffusion uniformity, coercivity, and the distribution of grain boundary phases, ultimately aiming to optimize the utilization of dysprosium in these alloys.

 I consider the topic to be both original and relevant to the field. The study addresses a specific gap in understanding how alloying elements can optimize the properties of rare earth metal-based materials. The research provides valuable insights that could enhance the efficiency and performance of these alloys in magnetic applications specially using NdFeB rare earth permanent magnets, making it a significant contribution to the field.

 This study adds new insights into how cerium enhances the diffusion behavior and magnetic properties of Dy-Cu alloys, specifically by lowering melting points and optimizing grain boundary phases. It fills a gap in existing literature by exploring cerium role at various concentrations and its influence on the diffusion behavior of the alloys for improving the utilization of dysprosium in advanced materials.

 The manuscript addresses an important concept in the study of Dy-Cu alloys; however, the authors should consider several improvements to their methodology and future work. They should include control experiments without cerium to better isolate its effects and explore a broader range of cerium concentrations for a more comprehensive analysis. Additionally, focusing on microstructural studies and changes would provide valuable insights and strengthen the study's findings.

The conclusions are consistent with the evidence and arguments presented. The study clearly demonstrates the influence of adding cerium on the diffusion behavior and properties of Dy-Cu alloys, effectively addressing the main question posed. The findings align well with the data, providing a solid basis for the conclusions drawn.

 The references are appropriate. They include relevant and recent studies that support the research context and findings.

 The tables and figures are generally well-constructed and effectively support the text and arguments made in the manuscript. They clearly present key data, enhancing the reader's understanding of the research findings.

Author Response

Reviewer 2: The manuscript “Reduction of Heavy Rare Earth Diffusion Sources in Sintered Nd-Fe-B Magnets via Grain Boundary Diffusion of Dy70Ce70-xCu30” provides important scientific results in the field of magnetic materials, particularly on the diffusion behavior of Dy-Cu alloys with Ce addition. It incorporates all relevant references and provides detailed reasoning and characterization.

However, upon reviewing the manuscript, I found some format editing is required. Please proofread and check the format and subscripts of the composition, for example, on page 3 in the last paragraph, and ensure the font size and subscripts of the compound composition are consistent. Adjusting these elements will improve readability and the overall quality of the manuscript.

Response: Thank you for your suggestion. The relevant format problems have been revised in the manuscript.

The research addresses how the addition of cerium (Ce) influences the diffusion behavior and overall properties of dysprosium-copper (Dy-Cu) alloys. The study specifically explores how Ce affects melting points, diffusion uniformity, coercivity, and the distribution of grain boundary phases, ultimately aiming to optimize the utilization of dysprosium in these alloys.I consider the topic to be both original and relevant to the field. The study addresses a specific gap in understanding how alloying elements can optimize the properties of rare earth metal-based materials. The research provides valuable insights that could enhance the efficiency and performance of these alloys in magnetic applications specially using NdFeB rare earth permanent magnets, making it a significant contribution to the field.This study adds new insights into how cerium enhances the diffusion behavior and magnetic properties of Dy-Cu alloys, specifically by lowering melting points and optimizing grain boundary phases. It fills a gap in existing literature by exploring cerium role at various concentrations and its influence on the diffusion behavior of the alloys for improving the utilization of dysprosium in advanced materials.

Response: Thank you very much for your professional guidance and suggestions, we hope to be able to conduct more valuable and meaningful research with your help.

The manuscript addresses an important concept in the study of Dy-Cu alloys; however, the authors should consider several improvements to their methodology and future work. They should include control experiments without cerium to better isolate its effects and explore a broader range of cerium concentrations for a more comprehensive analysis. Additionally, focusing on microstructural studies and changes would provide valuable insights and strengthen the study's findings.

Response: Your suggestions are highly valuable. We have already included experiments without Ce diffusion sources in the existing manuscript, which revealed a slightly lower performance enhancement compared to cases with Ce diffusion sources. Additionally, we plan to explore a broader range of cerium concentrations in our future research to conduct a more comprehensive analysis.

The conclusions are consistent with the evidence and arguments presented. The study clearly demonstrates the influence of adding cerium on the diffusion behavior and properties of Dy-Cu alloys, effectively addressing the main question posed. The findings align well with the data, providing a solid basis for the conclusions drawn.The references are appropriate. They include relevant and recent studies that support the research context and findings.The tables and figures are generally well-constructed and effectively support the text and arguments made in the manuscript. They clearly present key data, enhancing the reader's understanding of the research findings.

Response: Thank you very much for your acknowledgment and valuable feedback. We are delighted that you found the conclusions of the study consistent with the evidence and arguments presented, as well as the clear demonstration of the impact of adding cerium on the diffusion behavior and properties of Dy-Cu alloys. Your positive feedback is greatly appreciated. We will continue to ensure the appropriateness of references and strive to construct tables and figures that contribute to the reader's understanding. Once again, thank you for your valuable input.

Reviewer 3 Report

Comments and Suggestions for Authors

The study highlights the role of cerium (Ce) in enhancing the diffusion behaviour of Dy-Cu alloys, leading to improved magnetic properties. By optimizing the grain boundary phase structure and promoting uniform diffusion, the authors found that the right balance of Ce can significantly enhance the utilization of dysprosium (Dy) while maintaining the integrity of the magnet's crystal structure. The article also addresses the critical issue of heavy rare earth element (HREEs) usage in the production of Nd-Fe-B magnets, which are essential for various applications due to their high magnetic performance.

The authors employed a systematic approach to investigate the effects of Ce on the diffusion behaviour of Dy-Cu alloys. They utilized a commercial sintered magnet as the diffusion source and described the preparation of substrates through a series of defined processes (i.e. melting, rapid solidification and sintering). This methodological rigor ensures reproducibility and reliability of results. However, the article could benefit from a more detailed description of the experimental conditions, such as temperature profiles and specific measurements taken during the diffusion process.

The findings indicate that an appropriate amount of Ce can improve the crystal structure and magnetic properties of the magnets, while excessive Ce leads to distortion and degradation of these properties. The authors also reference previous studies that support the notion that Ce can enhance coercivity and reduce costs when used judiciously. The results provide a pathway for optimizing the use of rare earth elements in magnet production, which is crucial given the rising costs and supply chain concerns associated with them.

The discussion effectively synthesizes the results with existing literature, noting the complex interplay between Ce and other elements like Dy and Tb during the diffusion process. The authors acknowledge the need for further exploration of these interactions, which is a critical point for future research. However, the article could enhance its impact by including more quantitative data or comparative analyses with other diffusion sources to substantiate the claims made.

The article concludes that replacing HREEs with Ce in diffusion sources can lead to improved coercivity and reduced manufacturing costs. This conclusion is well-supported by the findings and aligns with the broader context of sustainable practices in material science. The authors provide a solid foundation for future research directions, particularly in exploring the synergistic effects of Ce with other elements.

The references cited are relevant and provide a good background for the claims made in the article. However, the inclusion of more recent studies could strengthen the literature review and provide a more comprehensive understanding of the current state of research in this area.

Overall, the article presents a solid contribution to the field of magnetic materials, particularly in the context of reducing reliance on heavy rare earth elements. The research is well-structured, and the findings are significant for both academic and industrial applications. Future studies should focus on the detailed mechanisms of Ce's interaction with other elements and the long-term stability of the modified magnets. Finally, by enhancing the methodological rigor (include more specifics on experimental conditions and measurements), providing quantitative data to support claims about magnetic property improvements, expanding the discussion, and updating the literature review (incorporating more recent studies to enhance the literature review), the authors can significantly strengthen their article and its contributions to the field.

Author Response

Reviewer 3: The study highlights the role of cerium (Ce) in enhancing the diffusion behaviour of Dy-Cu alloys, leading to improved magnetic properties. By optimizing the grain boundary phase structure and promoting uniform diffusion, the authors found that the right balance of Ce can significantly enhance the utilization of dysprosium (Dy) while maintaining the integrity of the magnet's crystal structure. The article also addresses the critical issue of heavy rare earth element (HREEs) usage in the production of Nd-Fe-B magnets, which are essential for various applications due to their high magnetic performance.

The authors employed a systematic approach to investigate the effects of Ce on the diffusion behaviour of Dy-Cu alloys. They utilized a commercial sintered magnet as the diffusion source and described the preparation of substrates through a series of defined processes (i.e. melting, rapid solidification and sintering). This methodological rigor ensures reproducibility and reliability of results. However, the article could benefit from a more detailed description of the experimental conditions, such as temperature profiles and specific measurements taken during the diffusion process.

The findings indicate that an appropriate amount of Ce can improve the crystal structure and magnetic properties of the magnets, while excessive Ce leads to distortion and degradation of these properties. The authors also reference previous studies that support the notion that Ce can enhance coercivity and reduce costs when used judiciously. The results provide a pathway for optimizing the use of rare earth elements in magnet production, which is crucial given the rising costs and supply chain concerns associated with them.

The discussion effectively synthesizes the results with existing literature, noting the complex interplay between Ce and other elements like Dy and Tb during the diffusion process. The authors acknowledge the need for further exploration of these interactions, which is a critical point for future research. However, the article could enhance its impact by including more quantitative data or comparative analyses with other diffusion sources to substantiate the claims made.

The article concludes that replacing HREEs with Ce in diffusion sources can lead to improved coercivity and reduced manufacturing costs. This conclusion is well-supported by the findings and aligns with the broader context of sustainable practices in material science. The authors provide a solid foundation for future research directions, particularly in exploring the synergistic effects of Ce with other elements.

The references cited are relevant and provide a good background for the claims made in the article. However, the inclusion of more recent studies could strengthen the literature review and provide a more comprehensive understanding of the current state of research in this area.

Overall, the article presents a solid contribution to the field of magnetic materials, particularly in the context of reducing reliance on heavy rare earth elements. The research is well-structured, and the findings are significant for both academic and industrial applications. Future studies should focus on the detailed mechanisms of Ce's interaction with other elements and the long-term stability of the modified magnets. Finally, by enhancing the methodological rigor (include more specifics on experimental conditions and measurements), providing quantitative data to support claims about magnetic property improvements, expanding the discussion, and updating the literature review (incorporating more recent studies to enhance the literature review), the authors can significantly strengthen their article and its contributions to the field.

Response: Thank you very much for your thorough review of our research and valuable feedback. Your comments are crucial for further refining our study. We are pleased that you appreciated our insights on the role of cerium in the diffusion behavior of Dy-Cu alloys and its relevance to Nd-Fe-B magnet production. Your suggestions regarding providing more detailed descriptions of experimental conditions, incorporating quantitative data to support claims of magnetic property improvements, expanding discussions, and updating the literature review will help strengthen our research findings. We will carefully consider your feedback and incorporate it into our future studies and revisions of the manuscript. Once again, we sincerely appreciate your review and support.
